# A Mentor-Led Text-Messaging Intervention Increases Intake of Fruits and Vegetables and Goal Setting for Healthier Dietary Consumption among Rural Adolescents in Kentucky and North Carolina, 2017

**DOI:** 10.3390/nu11030593

**Published:** 2019-03-11

**Authors:** Alison Gustafson, Stephanie B. Jilcott Pitts, Kristen McQuerry, Oyinlola Babtunde, Janet Mullins

**Affiliations:** 1Department of Dietetics and Human Nutrition, University of Kentucky, 206G Funkhouser Building, Lexington, KY 40506, USA; 2Department of Public Health, Brody School of Medicine, East Carolina University, 115 Heart Drive, Mailstop 660, Room 2239, Greenville, NC 27834, USA; jilcotts@ecu.edu; 3Applied Statistics Lab, University of Kentucky, 305B MDS Building, Lexington, KY 40506, USA; Kristen.mcquerry@uky.edu; 4Department of Nutrition Science, College of Allied Health Sciences, East Carolina University, Health Sciences Bldg. Rm. 2437, MailStop #668, 600 Moye Blvd., Greenville, NC 27834, USA; babatundeo@ecu.edu; 5Department of Dietetics and Human Nutrition, University of Kentucky, 206J Funkhouser Building, Lexington, KY 40506, USA; Janet.mullins@uky.edu

**Keywords:** text message, intervention, adolescent, fruit and vegetable intake

## Abstract

Introduction—Text-messaging interventions hold promise for successful weight loss interventions. However, there is limited research on text-messaging interventions to improve dietary intake among rural adolescents, who are at greater risk for obesity and related risk factors. The goal of this study was to test an eight-week, mentor-led text-messaging intervention among 14–16-year-old rural adolescents: the “Go Big and Bring It Home” Project to improve fruit and vegetable and healthy beverage intake. Methods and Materials—Eight rural high schools in eastern Kentucky and eastern North Carolina participated (*n* = 4 were randomized as intervention schools and *n* = 4 were randomized as control schools). Adolescents were recruited to participate in the eight-week text-messaging intervention. The text messages were primarily affective messages, and included a weekly challenge related to consuming fruits, vegetables, or healthy/low-calorie beverages. Undergraduate nutrition students sent text messages on Tuesday and Saturday every week over the eight-week period via the “Group Me” mobile application. Delayed controls received no information or text messages during the eight-week intervention. Fruit and vegetable intake was measured with the National Cancer Institute Fruit and Vegetable screener and beverage intake was assessed using the Beverage Questionnaire-10 (BEVQ-10). Intention-to-treat analyses were conducted among all those that completed the baseline and post-intervention survey (*n* = 277 intervention students and *n* = 134 delayed control students). All linear regression models were adjusted for race and were clustered on school to control for intraclass correlation. Results—In adjusted analyses, there was a statistically significant positive intervention effect on the primary outcome of fruit and vegetable servings/day with a mean difference between intervention and control participants of 1.28 servings/day (95% Confidence Interval 1.11, 1.48). There was no intervention effect on beverage intake. There was a statistically significant increase in the odds of goal setting for healthier dietary behaviors among intervention participants relative to controls. Conclusion—An eight-week text-messaging intervention led to increases in self-reported fruit and vegetable intake and improvements in goal setting for healthier dietary behaviors. Due to the use of undergraduate students to deliver the messages, and use of an existing web application, this text-messaging intervention can be sustained in underserved, rural environments. Thus there is potential for significant reach and public health impact to improve dietary patterns.

## 1. Introduction

Rural adolescents have higher rates of obesity and consume fewer fruits and vegetables relative to their urban counterparts [1,2]. Additionally, rural adolescents are at greater risk for cardiovascular disease and associated comorbidities relative to their urban counterparts [3]. A host of factors related to geographic isolation, socio-economic status, and lack of access to affordable healthy foods all contribute to the growing prevalence of obesity and poor dietary outcomes in rural communities [1,4].

Research has found that adolescents who frequent fast-food restaurants consume more added sugars and more calories from sugar-sweetened beverages [5], while those who frequently shop at gas stations and convenience stores consume more energy and servings of sweets and snack items [6]. In addition, adolescents who report greater availability of sugar-sweetened beverages (SSB) in their home have higher body weights and report greater intake of calories from SSB [7,8,9]. Furthermore, adolescents from households with greater availability of fruits and vegetables report higher intake of fruits and vegetables compared to those with lower availability [10]. Therefore, it is important to encourage adolescents to be advocates for healthy food in the home food environment and to have the skills to make healthy food purchases when they are in their communities.

In recent years, there has been increasing proliferation of interventions utilizing text messaging and web-based applications (“apps”) as tools to encourage and promote healthy behaviors. Such technology-based interventions have been successful for tobacco cessation, reducing alcohol abuse [11,12,13], and weight loss among adolescents [14]. Specifically related to dietary intake, several weight loss interventions have utilized mobile applications to deliver text messages containing tailored dietary feedback [15,16]. Overall, such strategies are effective at producing weight loss among adults [17], and improving purchases of fruits and vegetables and healthier snacks among urban adolescents [18]. An additional review related to decreasing sedentary behavior utilizing a text message approach with adolescents recommends including tailored feedback, goal setting, and sending at least two–three text message per week [19]. Yet, there are no interventions aimed at providing affective messages targeting intake of fruit, vegetable, and healthy/low-calorie beverages, specifically among rural-dwelling adolescents [20,21]. Moreover, there are limited interventions explicitly utilizing theory-based design in the text message content [21,22]. Uniquely tailored tone and structure of text messages improve receptivity, memory of content, and relevance of the message [23]. In addition, affective messages (focused on emotion) rather than instrumental messages (focused on knowledge and facts) increase receptivity of messages and lead to improved outcomes [16,20]. However, none have examined whether text message delivery improves key theoretical constructs such as self-efficacy and goal setting related to dietary intake.

Given the potential effectiveness of text messages for improving dietary intake among adolescents, we developed and tested a text-messaging intervention to increase self-efficacy and goal setting for healthier food and beverage intake. Messages included a weekly dietary challenge, encouraged adolescents to make healthier choices in the community, school, and home food environments, and focused on the adolescent as the agent of change in the household as a way to improve healthy food and beverage availability. The overall aims of this paper are to report the intervention effect on: (1) fruit, vegetable, and sugar-sweetened beverage intake compared to controls; (2) food shopping habits; (3) home availability of fruits, vegetables, healthy and less healthy snacks, and sugar-sweetened beverages; and (4) key theoretical constructs that were embedded within the messages (self-efficacy and goal setting).

## 2. Materials and Methods

### 2.1. Recruitment and Randomization of Included High Schools

A total of eight high schools (four in rural eastern Kentucky and four in rural eastern North Carolina) agreed to participate in the intervention in the fall of 2017. Schools were asked to participate in the intervention through Cooperative Extension agents in each county in Kentucky and in North Carolina through existing relationships with school staff and administration. A formative evaluation survey was completed among students, and results are reported elsewhere [24]. Results were used to inform the design of the Go Big and Bring it Home (GBBH) intervention content. The randomized controlled trial Clinical Trial Registration Number NCT02793024 began in fall 2017. Declarations: Ethical Approval and Consent to participate—University of Kentucky Institutional Review Board (IRB) approved this study under 16-0114-P4S and the ECU Institutional Review Board deferred to the UK IRB as the IRB of record. All adolescents completed assent forms and parents/guardians provided consent. Availability of Data and Materials: Data and all study materials will be provided to those submitting reasonable requests to Dr. Alison Gustafson Alison.gustafson@uky.edu or 859-257-1309. Materials will be made available online at www.uky.edu/HES.

### 2.2. Recruitment and Enrollment of High School Students

Advertising for recruitment was conducted through several channels. Students who completed a formative assessment survey in fall of 2016 [24] and indicated they were interested in being contacted for future studies were sent an e-mail and text message, and were given information sheets about the intervention (*n* = 310). Each high school had staff who assisted with distributing enrollment sheets to students and posted information on the school websites and/or Facebook web page. Several high schools had orientation events where students were given information sheets. Teachers handed out information to students in foods/culinary classes, physical education and health classes, home room, English classes, and in a general agriculture course.

Students were enrolled by providing contact information on an enrollment form, and were instructed to like the GBBH Facebook page and to download the Group Me mobile application if they had a cellular telephone. If students did not have a cellular telephone, they received messages via the Group Me app through their e-mail. (Only one student received messages in this manner). Students also had to return the informed consent from their parents and signed assent. After students completed the consent form, assent form, and baseline survey, high schools were randomized to either receive the intervention (round 1, beginning 30 September 2017) or delayed control intervention (round 2, March 2018). In Figure 1 a graphical depiction of student enrollment, drop-outs and completers is presented. A total of 530 students completed all necessary information to be enrolled in the study. Of those 530, 150 were from control schools and 380 were from intervention schools. After students were invited to join the Group Me app to receive text messages, 48 students from the intervention arm dropped from the study. The final enrollment for the intervention group was *n* = 332. During the course of the intervention, a large proportion of the students never responded to any text messages. These students (*n* = 137, 41.3%) were defined as “nonresponders”. (For the purposes of this report, we conducted intention-to-treat analyses, comparing responses of both “responders” and “nonresponders” (all in the intervention group) to control group participants.)

At baseline of enrollment, students answered a 30-minute survey about food shopping behaviors, questions related to the core constructs, dietary intake, and demographics. The students received $25 for participating in this baseline survey. Postintervention, the same survey was conducted to assess change in key behavioral outcomes and theoretical constructs. Intervention and control students received $30 for participating in the postintervention survey. A total of *n* = 411 completed the post-intervention survey (*n* = 134 control and *n* = 277 intervention).

### 2.3. “Go Big and Bring it Home” (GBBH) Intervention Components

Undergraduate students in human nutrition and dietetics from the University of Kentucky were recruited through an e-mail listserv and asked to volunteer as mentors for enrolled GBBH participants. Undergraduate volunteers participated in an hour-long training about how to effectively send text messages as the mentor via the Group Me app. A total of 34 undergraduate students were enrolled as mentors. Of the mentors, 32 were females and 2 were males. These mentors were supervised by four Master of Science/Registered Dietitian graduate students at the University of Kentucky.

A total of eight weeks of text messages were sent, every week on Tuesday (after 4 pm so as not to disrupt the school day) and Saturday (between 12 and 2 pm). There was a warm-up introduction week, where students got to know their mentor and work out any communication glitches. The next six weeks covered nutrition-related content. The summary week congratulated students and provided information about the goals that individual students and schools achieved.

Data from the formative survey in fall 2016 [24] and evidence from the empirical literature were used to develop intervention content. Text message content each week focused on a different food venue where the adolescents reported purchasing or consuming food (convenience stores; supermarkets; fast-food restaurants) in the formative survey. Examples of messages are as follows: “Choose a fruit when shopping at the gas station”; or “I like to grab water when I eat fast food, think that might work for you this week?” The text messages were affective in content and tone, to encourage adolescents to purchase fruits, vegetables, and lower-calorie beverages [20]. Prior research indicates that adolescents want positive messages about behaviors, and do not want to hear negative messages about foods and beverages to avoid [14]. Therefore, the text messages did not include negative wording such as “no” or “stay away from”. The text messages alternated over the six core weeks between fruit, vegetable, and low-calorie beverages messages. Additionally, there was a focus on shopping with parents and how to improve the home food environment, based on previous findings that several healthy changes could be made at home [24]. The core constructs of self-efficacy and goal setting for purchasing and consuming more fruits, vegetables, and low- or no-calorie beverages were embedded within text messages. These constructs were grounded in social cognitive theory and a review of the literature [12,25,26]. All message templates and response options were developed by the Principal Investigator with review from undergraduate students for tone and length acceptability for messages to be sent via text.

The first text message sent each week was a standardized script often containing a question related to the study goals, in order to prompt conversation (e.g., “Have you eaten a fruit yet today?”) sent to each high school student. After the first message was sent, text messages were tailored after the high school student responded with “got it” (indicating that he/she had achieved the goal) or “not yet” (indicating the goal or challenge had not yet been achieved). The text message was a “two-way” communication between the undergraduate student and the high school students, meaning that messages were sent back and forth so communication was established between the peer mentor and adolescents. Based on how the student responded, the undergraduate would offer encouragement and assistance with goal setting, and try to improve the self-efficacy of the adolescent.

Students in the intervention group were provided a $5 cash incentive each week if they returned the text message on Tuesday and Saturday. The undergraduate students also received gift cards for participating in the program (*n* = 34).

In order to monitor text messages and content, four graduate students and the PI of the study monitored each text message delivery sent between the undergraduate and the high school student each week. If the undergraduate student missed sending a text message challenge to the participating high school students, the graduate student sent the message within the same day. This occurred for two of the same undergraduate students approximately 12 times over the course of the intervention.

### 2.4. Dependent Variables

The primary outcome of interest was fruit and vegetable intake, as assessed via the National Cancer Institute (NCI) Fruit and Vegetable Screener [27]. Fruit and vegetable servings/day were calculated based upon NCI’s standard scoring algorithms found at https://epi.grants.cancer.gov.diet/screeners/fruitveg/scoring/allday.html. Each reported frequency was converted to an average daily frequency, multiplied by the number of MyPyramid servings for the portion size, and summed across all food groups.

Secondary outcomes of interest were sugar-sweetened beverage intake, Body Mass Index (BMI) *Z*-score, home food availability, purchasing habits, self-efficacy, and goal setting related to healthy eating. Beverage consumption was assessed from the validated beverage questionnaire (BEV-Q10), wherein participants are asked about consumption frequency for several different beverages. Beverages included sweetened juice, soft drinks (regular and diet), sweetened tea, energy drinks, 100% fruit juice, and milk (whole, 2%, skim). Coding was based on the standard protocol developed and validated by Hedricks et al. [28]. Briefly, for each beverage, the amount of each beverage an individual consumed was multiplied by how often the beverage was consumed to quantify the average daily fluid ounces (fl oz). The average fl oz was then multiplied by the average kilocalories per fluid ounce to attain the average daily kilocalories. Summing together the average daily calories produced the total average daily beverage calorie intake. The average fl oz variable was also used to calculate the average daily grams by multiplying it by the average g/fl oz. Similarly, the summed average daily grams produced the total average daily beverage grams.

BMI *Z*-score was calculated from self-reported height and weight. Directions for Z-score calculations came from the Centers for Disease Control and Prevention (CDC, www.cdc.gov). Median, generalized coefficient of variation, and the power in the Box–Cox transformation were provided by the CDC. *Z*-scores for each individual were derived by taking the weight and dividing by the median and raising that quantity to the Box–Cox transformation value. This quantity was subtracted by 1 and then divided by the product of the Box–Cox transformation and the generalized coefficient of variation. The corresponding percentile to the calculated *Z*-score was obtained from a standard normal distribution.

Validated questions from the Project EAT Study (University of Minnesota) were used to ascertain availability of food within the home [21,22,23]. Adolescents were asked whether certain food and beverage categories were available in the home (i.e., “Fruits and vegetables are available in my home”, “We have ‘junk food’ in my home”, and “Soda pop is available in my home.”) with responses ranging from “never” to “always”. Based on the distribution of the data, the home food environment categories were collapsed into the following categories: never available, sometimes available, and always available.

As used in a prior study [29], participants were asked where they purchased foods from the following food venues: supermarkets, corner stores/convenience/gas stations, dollar stores, fast-food restaurants, school cafeteria. Due to the distribution of data, we only report on shopping habits at supermarkets, gas stations, and fast-food restaurants. The survey ascertained if a food or beverage item was purchased at each food venue. Food items were classified into the following categories to mimic the dietary suggestions made in the text messages: (1) Fruits and vegetables—apples, bananas, green salad, etc.; (2) Less-healthy snacks—chips, baked goods (cookies, cakes, donuts), chocolate, sugar candy, ice cream, snow balls, popsicles; (3) Healthy snacks—baked chips, reduced-fat chips, pretzels, nuts, seeds, yogurt, granola bars; (4) Sugar-sweetened beverages—regular soda, sweet tea, fruit punch, sports drink, energy drink; and (5) Water or low-calorie beverages—water, flavored water, unsweetened tea.

Self-efficacy was assessed using questions validated in an adolescent population [26], including: “I know I can eat vegetables several times a day”; “I know I can substitute veggies for chips as a snack if I try hard enough”. The response options consisted of “I know I can”, “I think I can”, “I’m not sure I can”, and “I know I can’t”. A binary variable was created for high self-efficacy, which included responses of “I know I can” and “I think I can”, and low self-efficacy, which included responses of “I’m not sure I can” and “I know I can’t”. To create the self-efficacy variables specific to fruit (self-efficacy fruit), vegetable (self-efficacy vegetable), and healthy snacks (self-efficacy snacks) consumption, Spearman rank correlation was conducted across all eight questions.

Goal setting was assessed with the following questions: “How often do you set a goal for yourself related to eating fruit?”; “How often do you set a goal for yourself related to eating vegetables?”; “How often do you set a goal for yourself to drink beverages without sugar (including artificially sweetened beverages like Diet Coke)?” The response options included “almost never”, “sometimes”, “often”, and “almost always”. A binary variable was created for high goal setting including responses of “often” and “almost always” as one category for “high” and low goal setting including the responses of “sometimes” and “almost never” as one category for “low”.

Baseline demographic variables were examined using means (standard deviations) and frequencies (percentages). Pre- and post-values and mean changes within the intervention and control groups were examined using t-tests. Analyses also examined differences between control students, responder students, and non-responder students in the intervention on variables of interest. There were no statistically significant differences (results available upon request). To determine the change among primary and secondary outcomes between intervention compared to controls, linear regression was used, adjusted for race since randomization did not result in equal distributions of racial groups in intervention versus control groups. Additional covariates were baseline values of primary and secondary outcomes with a cluster command on school since students were nested within schools and to provide robust standard errors. Primary and secondary outcomes were log-transformed based on the Shapiro Wilk test for normality. After log transformation, the *p*-value was 0.05 and could be rejected and thus analyses was done with log-transformed variables. Results are presented back-transformed for ease of interpretation.

To assess percent changes on home availability, the Stuart Maxwell test was used. To assess changes in key constructs of self-efficacy and goal setting, logistic odds ratios were used, controlling for race and cluster command on school. Analyses were conducted using intention-to-treat. All analyses were conducted in Stata 14.0 [30].

## 3. Results

Table 1 provides information on baseline demographics and characteristics of the intervention and control participants. The mean age was 15 years for both control and intervention sites. There were more females compared to males enrolled in the GBBH intervention among the entire study sample. There were baseline statistical differences between control and intervention participants for race/ethnicity, and servings of fruit and vegetable intake. Otherwise, control and intervention adolescents were not significantly different on any other variables. Additionally, both intervention and control adolescents purchased a greater percentage of SSB from fast-food restaurants and convenience stores compared to the percentage purchasing water or no-calorie drinks at these types of venues. Among the full sample of intervention and control participants, 27% reported buying SSB at fast-food restaurants, while 15% and 14% of intervention and control participants, respectively, reported buying water or no-calorie beverages at fast-food restaurants. A high percentage of all food categories were purchased at supermarkets relative to the other food venues.

Table 2 provides information on the mean changes within and between the intervention and control groups for primary and secondary outcomes. There was a significant and positive intervention effect on intake of fruit and vegetable servings/day. The control participants reported a decrease of 1.52 servings/day, while intervention participants reported an increase of 0.71 daily fruit and vegetable servings, however, results were not significant. There was a significant mean effect between intervention and control of 1.28 (95% CI 1.11, 1.48) servings of fruit and vegetables/day. There was no intervention effect on SSB consumption. There was an increase in fruit and vegetables purchases over 7 days (2.55 purchases/week, 95% CI 0.69, 4.42); healthy snack purchases (1.81 snacks/day 95% CI 0.68, 2.94); and water or no-calorie beverages (0.87 calories/day 95% CI 0.18, 1.56) within intervention adolescents. There was also an increase in healthy snack purchases (1.87 snacks/day, 95% CI 0.54, 3.20) among control participants.

The intervention messages also targeted improving the home food environment. There was a significant change within the intervention group for junk food (*p* = 0.008), with an increase of 8% on junk food never being available in the home and a decrease of −4% and −5% of junk food sometimes or always being available, respectively. Among control participants, there was a −13% decrease of fruits and vegetables always being available in the home, 6% increase in sometimes, but a 7% increase in never having fruits and vegetables being available in the home.

Table 3 provides information on changes in self-efficacy and goal setting. Those in the intervention group reported higher odds of having high self-efficacy for eating vegetables compared to controls (1.59 OR, 95% CI 1.19, 2.13). The goal setting construct indicates that those in the intervention reported higher odds of goal setting for each dietary goal compared to controls (Fruit—1.52 OR 95% CI 1.18, 1.95); (Vegetable—1.75 OR 95% CI 1.19, 2.58); and (Sugar-free beverage—1.94 OR 95% CI 1.18, 3.27).

## 4. Discussion

A mentor-led text message intervention was effective at increasing self-reported fruit and vegetable intake among rural adolescents. This intervention was also effective at improving several key secondary outcomes related to purchasing habits, home availability, self-efficacy for vegetable consumption, and goal setting for healthier dietary choices. Of note is the decrease in fruit and vegetable intake among control students, which led to the intervention effect between control and intervention adolescents. Given the time frame of pre- and post-data collection, this finding of a decrease among controls is found within the literature [31,32], such that the lowest intake of fruits and vegetables is found among adolescents. Although the GBBH intervention adolescents did not significantly increase fruit and vegetable intake, they did not decrease intake, and thus, the intervention may prove a useful tool in improving dietary intake over time. Previous studies have reported that affective messages are effective to increase fruit and vegetable intake among adolescents [18,20]. Our results corroborate these findings among adolescents from two rural communities. Thus, text-messaging holds promise as a delivery method that can be effective across a variety of settings as a way to improve adolescents’ dietary intake.

Results indicate that adolescents within the intervention schools reported improvements in the types of foods and beverages purchased, although not significant relative to controls. However, within the intervention, there was a significant effect in water or no-calorie beverages being purchased. While not statistically significant relative to controls, these findings hold promise for future interventions aiming to improve shopping behaviors among adolescents, a critical priority population for behavior change as adolescents are transitioning into adulthood. To facilitate healthier purchases in future studies, intervention content could use ecological momentary assessment (prompts that are capturing where the individual is shopping at the given moment [33]) techniques to send messages when adolescents are “in the moment” of purchasing food at various food venues.

Text messages also encouraged adolescents to be the agent of change for healthier food within the home. Results indicate that there was a significant decrease in the percentage of intervention adolescents who reported “junk food” was available in the home. While the study did not explicitly teach what foods were considered “junk”, the text messages advocated for healthy alternatives over foods higher in processed sugars. Focusing on food that is made available in the home could have a profound influence on adolescents’ dietary behaviors, as food available in the home is associated with dietary intake in prior studies [9,34].

Lastly, the intervention employed explicit language targeting self-efficacy and goal setting within each message. Our results illustrate how targeting these key constructs can help to improve goal setting and self-efficacy. These constructs are highly relevant for improving dietary intake, as previously reported by others [21,23,35,36,37]. Thus, our current findings provide further support for the need for text-messaging interventions to utilize a theory-based approach for delivery.

There are key limitations of this intervention. Although every attempt was made to recruit and enroll adolescents, there is a strong possibility that those who were already eating healthy chose to participate in this trial. However, since schools were randomized, this limits self-selection bias between intervention and control participants. The racial composition of the participants mirrors the geographic area of where they reside, yet all the minority adolescents resided in North Carolina. Our analyses did adjust for race and ethnicity, but future work should focus on obtaining a more representative sample. This study was limited in geographic scope and should be replicated among a larger sample across more states. One student received messages via e-mail, limiting the immediacy and “real-time” effects that a text message can offer, which most likely limited this participant’s intervention dose. However, sensitivity analyses were conducted both with and without this student in the sample, and point estimates did not change. Finally, self-reported dietary intake was our primary outcome, and self-reported intake is subject to social desirability bias, intervention-related bias, and recall bias [38,39].

## 5. Conclusions

A mentor-led text-messaging intervention with affective messages that focused on a variety of individual- and environmental-level changes holds promise for improving dietary and purchasing behaviors and the home food environment among rural adolescents. Future work will focus on replicating this study for further dissemination among health departments, Cooperative Extension, and organizations working with adolescents to improve health outcomes.

## Figures and Tables

**Figure 1 nutrients-11-00593-f001:**
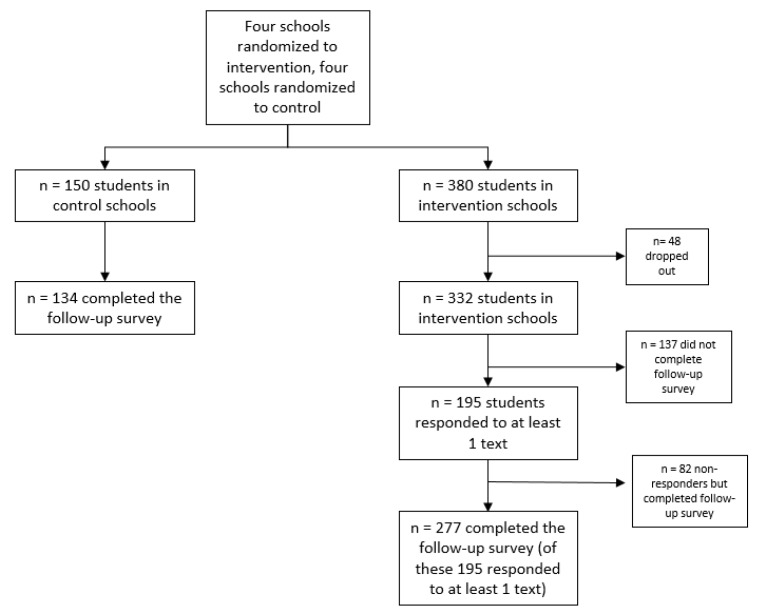
Student enrollment, drop-outs, and completers.

**Table 1 nutrients-11-00593-t001:** Demographics baseline characteristics of intervention and control participants (*n* = 411).

	Intervention (*n* = 277)	Control (*n* = 134)	
**Age (mean)**	15 (0.07)	15 (0.10)	*p* = 0.80
**Gender**			*p* = 0.70
Female	61%	67%	
Male	38%	30%	
Other (Male to Female; Female to Male; Other)			
**Race**			
White	72%	55%	*p* = 0.01
Other (African American/Hispanic)	28%	45%	*p* = 0.01
**Dietary Intake and Body Mass Index**			
Fruit and vegetable servings (mean and SE)	3.83 (0.24)	5.56 (0.54)	*p* = 0.01
Sugar-sweetened beverage calories (mean and SE)	439 (31)	470 (54)	*p* = 0.6
Sugar-sweetened beverage grams (mean and SE)	1112 (79)	1200 (140)	*p* = 0.6
BMI Z-score percentile (mean and SE)	0.76 (0.02)	0.72 (0.02)	*p* = 0.9
**Food Shopping Practices**			*p* = 0.60
Buy fruit and vegetables (mean times per week purchased)	16 (0.99)	13 (0.92)	
Buy fast food (mean times per week)	31 (0.99)	27 (1.26)	
Buy healthy snack foods (mean times per week)	8.74 (0.54)	6.39 (0.59)	
Buy less-healthy snack	15.88 (0.84)	13.58 (1.05)	
Buy SSB	10.33 (0.48)	9.95 (0.68)	
Buy water or no-calorie beverages	5.38 (0.30)	5.22 (0.40)	
**Where Food is Purchased**			*p* = 0.07
*Fruit and Vegetables*			
Supermarket	86%	98%	
Convenience Stores	13%	10%	
Fast Food	1%	4%	
*Sugar-Sweetened Beverages (regular soda; fruit punch; sports drink; energy drink; sweet tea)*			
Supermarket	83%	92%	
Convenience Stores	32%	24%	
Fast Food	27%	22%	
*Water or Low-Calorie Beverages (water; flavored water; unsweet tea)*			
Supermarket	78%	86%	
Convenience Stores	7%	7%	
Fast Food	15%	14%	
*Healthy Snacks (pretzels; low-fat chips; granola bars; nuts/seeds; yogurt)*			
Supermarket	87%	90%	
Convenience Stores	14%	13%	
Fast Food	6%	9%	
*Less-Healthy Snacks (full-calorie chips; cookies; cakes; donuts; chocolate candy; sugar candy; ice cream; popsicles; snow cones)*			
Supermarket	88%	90%	
Convenience Stores	28%	18%	
Fast Food	11%	12%	
**Home Availability (Always Available)**			
Fruit and Vegetable	45%	50%	
Vegetables Served at Dinner	39%	25%	
Junk Food	31%	31%	
Soda	36%	36%	
Snacks	27%	30%	
Candy	19%	18%	
**Core Constructs (I know I can)**			
*Self-Efficacy*			
Choose Fruit	63%	51%	
Choose Vegetables	47%	25%	
Choose Healthy Snacks	53%	42%	
*Goal Setting*			
Fruit Goal Almost Always	10%	10%	
Veggie Goal Almost Always	10%	8%	
Sugar-Free Beverage Goal	19%	12%	
**Level of Engagement**			
Text messages returned (mean total messages over eight weeks)	11.34 (0.73)	N/A	
Avid responder (returned at least two messages per week)	63%	N/A	

**Table 2 nutrients-11-00593-t002:** Intervention effect of mentor-led text message on dietary intake, food shopping practices, and home availability.

Dietary Intake and BMI	Change within Intervention Participants	Change within Control Participants	Difference between Intervention and Control Participants Postintervention
Fruit and vegetable servings/day	0.71 (−0.02, 1.45)	−1.52 (−2.48, −0.56) *	1.28 (1.11, 1.48) *
Sugar-sweetened beverage calories	−61 (−125, 80) *	−39 (−149, 71)	−39 (−164, 85)
Total beverage calories	−26 (−139, 87)	−118 (−320, 83)	14.99 (−186, 216)
Body Mass Index *Z*-score percentile	−0.005 (−0.02, 0.009)	0.002 (−0.01, 0.02)	0.99 (0.92, 1.07)
***Food shopping practices (number of times an item was purchased over 7 days)***			
Fruit and vegetable purchases over 7 days	2.55 (0.69, 4.42) *	1.39 (−0.65, 3.44)	1.07 (0.94, 1.22)
Less-healthy snack purchases over 7 days	0.82 (−0.88, 2.53)	−0.18 (−2.13, 1.75)	1.06 (0.85, 1.33)
Healthy snack purchases over 7 days	1.81 (0.68, 2.94) *	1.87 (0.54, 3.20) *	1.02 (0.84, 1.23)
SSB beverage purchases over 7 days	0.26 (−0.71, 1.24)	−0.25 (−1.64, 1.14)	1.02 (0.85, 1.21)
Water or no-calorie beverage purchases	**0.87 (0.18, 1.56) ***	0.36 (−0.58, 1.31)	1.03 (0.88, 1.21)
**Home Availability (Percent Change Reported)**			
*Fruit and vegetables*	*p* = 0.3	***p* = 0.02**	***p* = 0.03**
Never	<1%	7%	
Sometimes	4%	6%	
Always	5%	−13%	
*Vegetables served at dinner*	*p* = 0.08	*p* = 0.2	*p* = 0.03
Never	−2%	−12%	
Sometimes	7%	11%	
Always	−5%	−3%	
*Junk food*	*p* = 0.008	*p* = 0.68	*p* = 0.01
Never	8%	1%	
Sometimes	−4%	3%	
Always	−5%	−4%	
*Soda*	*p* = 0.07	*p* = 0.23	*p* = 0.05
Never	6%	<1%	
Sometimes	−3%	6%	
Always	−3%	−6%	

* *p* < 0.05. *t*-tests were conducted to compare mean changes within intervention and control group. Linear regression was conducted to test intervention effect on key outcomes controlling for baseline measures and race and clustering on school. Stuart Maxwell was used to assess percent changes for home availability between intervention and control participants.

**Table 3 nutrients-11-00593-t003:** Intervention effect on constructs embedded within the text messages.

	Odds Ratio 95% CI
Self-Efficacy	
Fruit	0.93 (0.73, 1.29)
Vegetable	1.59 (1.19, 2.13) *
Healthy Snacks	0.72 (0.48, 1.09)
Goal Setting	
Fruit	1.52 (1.18, 1.95) *
Vegetable	1.75 (1.19, 2.58) *
Sugar-Free Beverage	1.94 (1.18, 3.27) *

Self-efficacy reference is low self-efficacy and being a control. Goal-setting reference is low for goal setting and being a control. * = *p* < 0.05.

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
