# Peer review of "A Mentor-Led Text-Messaging Intervention Increases Intake of Fruits and Vegetables and Goal Setting for Healthier Dietary Consumption among Rural Adolescents in Kentucky and North Carolina, 2017"

_nutrients, 2019, doi:10.3390/nu11030593_

Round 1

Reviewer 1 Report

This work is well written and the intervention well performed. It shows a nice approach that might be useful for the treatment of obesity in American adolescents. I think it has a good number of cases, although the self-reporting by the adolescents might have some problems related to bias, as already commented by the authors. I have a few concerns and suggestions that I feel are necessary or that would improve this work:

1.       Line 137-138: You specify that one student received messages through his e-mail. Since he had to use a computer, he probably did not have the same immediacy that a cellular can offer, or the possibility to read the messages while on the supermarket. Do you think this can influence the effect that the messages had in this adolescent? Why was this student included in the data analysis?

2.  Line 305-307 and 319-320: How do you explain that fruit and vegetable consumption/availability decreased in the control group? I think this should be commented in the discussion.

3.       Line 307: You state that there was an increase of 0.71 daily fruit and vegetable servings; however, in the table this is shown as not significant. This should be stated in the text too.

4.       Line 310-312: You comment that there was an increase in healthy snack purchases in the intervention group, however, the same was true for the control group. These results should be commented both in the results and discussion section.

5.       Line 328-329: You state that the intervention increased fruit and vegetable intake; however, this is not shown in the Table 2 (.71 (-.02, 1.45)). There are no differences when comparing the pre and post values within the intervention. There are differences between intervention and control, however, this is caused by the (I suspect) unexpected decrease in Fruit and vegetable servings in the pre and post control data, and do not seem to happen by a direct increase in the intervention group. I think this should be commented in the discussion, since I think it is an important detail.

6.       Line 335-336: It might be appropriate to specify that although there are no differences compared to the control group, there were differences between the pre and post values in the intervention.

7.       Line 346: Were adolescents taught what exactly is “junk food”? It might be obvious to the majority of them, however it could be an important factor when self-reporting dietary intake.

8.       Minor comments:

a.       Be sure to always write servings/day or servings/ day, not both.

b.       The first table should be Table 1, then Table 2, …

Author Response

We thank the reviewer for this very thoughtful review of our manuscript. We have made all the suggested edits and changes and provide a point by point response below.

1.           Line 137-138: You specify that one student received messages through his e-mail. Since he had to use a computer, he probably did not have the same immediacy that a cellular can offer, or the possibility to read the messages while on the supermarket. Do you think this can influence the effect that the messages had in this adolescent? Why was this student included in the data analysis?

We think the reviewer raises a very strong point. This student remained in the analyses since removing the student did not change the point estimate. We have however added this key point to the limitations section of the manuscript. The following has now been inserted:

One student received messages via e-mail, limiting the immediacy and “real time” effects that a text message can offer, which most likely limited this participant’s intervention dose. However, sensitivity analyses were conducted both with and without this student in the sample, and  point estimates did not change.

2.           Line 305-307 and 319-320: How do you explain that fruit and vegetable consumption/availability decreased in the control group? I think this should be commented in the discussion.

We have added this point in the discussion with the following:

The decrease in fruit and vegetable intake among control students is noteworthy, and is the reason for the statistically significant intervention effect on fruit and vegetable intake between control and intervention adolescents. Given the time frame of pre and post data collection this finding of a decrease among controls is reasonable given prior literature which found that adolescent intake decreases over time31, 32. Thus, although the intervention adolescents did not significant increase fruit and vegetable intake, they did not decrease intake, as would be expected given prior literature. 31,32 In conclusion, the intervention may prove a useful tool in preventing the natural decrease in fruit and vegetable intake over time. Previous

3.           Line 307: You state that there was an increase of 0.71 daily fruit and vegetable servings; however, in the table this is shown as not significant. This should be stated in the text too.

We have added that this was not significant in the results section.

The control participants reported a decrease of 1.52 servings/day, while intervention participants reported an increase of 0.71 daily fruit and vegetable servings, however results were not significant.

4.           Line 310-312: You comment that there was an increase in healthy snack purchases in the intervention group, however, the same was true for the control group. These results should be commented both in the results and discussion section.

We have added the following in both the results and discussion section:

There was also an increase in healthy snack purchases (1.87 snacks/day, 95% CI 0.54, 3.20) among control participants.

5.           Line 328-329: You state that the intervention increased fruit and vegetable intake; however, this is not shown in the Table 2 (.71 (-.02, 1.45)). There are no differences when comparing the pre and post values within the intervention. There are differences between intervention and control, however, this is caused by the (I suspect) unexpected decrease in Fruit and vegetable servings in the pre and post control data, and do not seem to happen by a direct increase in the intervention group. I think this should be commented in the discussion, since I think it is an important detail.

We agree with the reviewer and thank you for raising this key point. We have added the following in the discussion section:

Of note, is the decrease in fruit and vegetable intake among control students, which led to the intervention effect between control and intervention adolescents. Given the time frame of pre and post data collection this finding of a decrease among controls is found within the literature31, 32. Such that, the lowest intake of fruits and vegetables is found among adolescents. What is key in this finding is that although the intervention adolescents did not significant improve their intake, they did not decrease which would be more typical in this population. Thus, the intervention may prove a useful tool in assisting adolescents to not decrease intake over time.

6.           Line 335-336: It might be appropriate to specify that although there are no differences compared to the control group, there were differences between the pre and post values in the intervention.

We have added the following:

However, within the intervention there was a significant effect in water or no calorie beverages being purchased.

7.           Line 346: Were adolescents taught what exactly is “junk food”? It might be obvious to the majority of them, however it could be an important factor when self-reporting dietary intake.

The messages did not explicitly teach “junk food” but rather focused on healthy alternatives. We have added the following:

While the study did not explicitly teach what foods were considered “junk food”, the messages advocated for healthy alternatives over foods higher in processed sugars. Focusing on food that is made available in the home could have a profound influence on adolescents’ dietary behaviors, as food available in the home is associated with dietary intake in prior studies9, 34.

8.       Minor comments:

a.       Be sure to always write servings/day or servings/ day, not both.

We have changed to servings/day in the results section and throughout.

b.       The first table should be Table 1, then Table 2, …

We regret the mistake on table order as noted by the other reviewer and have fixed our erro.

Reviewer #2

Thank you for your comments and time on the manuscript review. We have fixed the table order has suggested. We also appreciate the insight about qualitative studies in our future work.

Reviewer 2 Report

The main aim of the paper is to test a randomized controlled trial of an 8-week, peer-led text messaging intervention among 14-16-year-old rural adolescents.

The study is well conducted and the results are interesting. It could have been complemented with interviews with the students, but still is interesting.

Specific comments

Line 288. Table 3 is before Table 1.

Author Response

The study is well conducted and the results are interesting. It could have been complemented with interviews with the students, but still is interesting.

 Thank you for your enthusiasm about our manuscript. 

Specific comments

Line 288. Table 3 is before Table 1.

We apologize for the error and have fixed the tables.